# Techniques for the surgical correction of lagophthalmos secondary to leprosy: A systematic review

Matthew Willis[1,2]ʘ*, Heynes Brown[3]ʘ, Dan McGrath[3], Essam Eltoukhy[4], Anil Fastenau[1,2,5,6]

**1** Department of Global Health, Institute of Public Health and Nursing Research, University of Bremen, Bremen, Germany, **2** Marie Adelaide Leprosy Center (MALC), Karachi, Pakistan, **3** School of Medicine, Dentistry & Biomedical Sciences, Queen's University Belfast, Belfast, United Kingdom, **4** Cairo University, Cairo, Egypt, **5** German Leprosy and Tuberculosis Relief Association (GLRA/DAHW), Wurzburg, Germany, **6** Heidelberg Institute of Global Health (HIGH), Medical School, Heidelberg University, Heidelberg, Germany

ʘ These authors contributed equally to this work.
\* willis@uni-bremen.de

## Abstract

### Background

Leprosy is an infectious neglected tropical disease that can result in chronic immune mediated nerve damage. When this involves the facial nerve, this can lead to lagophthalmos, with the later stages requiring surgical correction. If untreated lagophthalmos can cause keratopathies leading to visual impairment and eventual blindness. However, to date no paper has systematically reviewed the surgical management of lagophthalmos in those affected by leprosy.

### Methods

A systematic review was conducted on the 16/11/2024 with data from PubMed, Infolep, Web of Sciences Core Collection and Medline ALL. Data extraction and analysis followed PRISMA guidelines. Included were English-language studies on the outcomes of surgical procedures for the surgical management of lagophthalmos, regardless the year of publication.

### Results

The 12 papers identified contained data from seven countries. The majority of papers studied Temporalis Muscle transfer. Gillies technique or modifications of this technique were reported in five papers. Three papers reported modifications of Johnson's method of temporalis tendon transfer. One paper reported TMT using a silicone sling. Patients also received TMT using the Brown-McDonnell and the McCord-Cordner techniques. Lagophthalmos was also corrected using gold or steel weight implant

**Data availability statement:** All relevant data are in the manuscript and its supporting information files.

**Funding:** The author(s) received no specific funding for this work.

**Competing interests:** The authors have declared that no competing interests exist.

techniques. One paper each studied lateral tarsal strip, modified tarsorrhaphy and scapha graft.

## Discussion/conclusion

Treatment of lagophthalmos is vital to preserve vision in those affected by leprosy, however, it is important to take into consideration the practical advantages of the five broad techniques identified by this review. Factors such as the type of anaesthesia, level of expertise, success rate, incidence and risk of complications, and longevity and stability of the results, are vital to consider when conducting these surgical procedures in reduced resource settings. Therefore, operations which are more cost effective, show a reduced complication rate and yield better long-term results without complicated follow-up are more likely to be adopted in lower resource settings.

### Author summary

Leprosy is a chronic infectious neglected tropical disease. Leprosy has a high burden of visual impairment if left untreated, with both cataracts and corneal opacification being the main contributors to ocular morbidity. Lagophthalmos in those affected by leprosy, an inability to close the eyelids fully, can lead to exposure keratitis and eventual corneal opacification and blindness. The main way to avoid and prevent the progression of visual impairment once lagophthalmos is established is corrective surgery. We identified 12 studies from seven countries researching five main techniques for surgical correction of lagophthalmos, with the majority of papers (seven) investigating methods of temporalis muscle transfer. With three papers studying weight implantation, and one paper each covering scapha grafting, modified tarsorrhaphy and lateral tarsal strip. Surgical techniques were mostly well tolerated with minimal side effects noted across techniques. Of note however was the high incidence of extrusion and local allergic reactions within the gold weight technique. Ultimately, this paper presents a compelling case for more comparative studies between different techniques for lagophthalmos correction secondary to leprosy. Our recommendations emphasise the need to evaluate these techniques further in lower-resources settings and to prioritize techniques that allow good outcome with the minimal need for follow-up.

## Introduction

Leprosy, also known as Hansen's disease, is a chronic infectious disease caused by *Mycobacterium leprae* and *Mycobacterium lepromatosis* [1]. Leprosy is transmitted when droplets from a person with untreated leprosy spread to susceptible individuals, the disease is not spread through casual contact such as sharing a meal together or hugging and requires months of prolonged contact for spread to occur. Furthermore,

the ability to spread the disease is stopped as soon as the patient begins multidrug therapy (MDT). The disease primarily affects the skin, nerves, eyes and mucous membranes. The resulting immune mediated nerve damage can lead to injuries such as cuts and burns due to the loss of sensation [2].

Ocular leprosy leads to many associated signs and symptoms in the ocular region, as well as in the bulb and adnexa (Eyelids, eyebrows and lacrimal system). Among these signs the most dangerous to vision are those of lagophthalmos and corneal hypoesthesia, leading to exposure and neurotrophic keratopathies respectively. Although Multidrug therapy interrupts the spread and "cures" patients of infections it does not remove the risk of ocular complications, and those with treated leprosy continue to experience high levels of ocular morbidity [3]. Today due to the success of MDT in limiting the advancement of neurological sequelae of the disease age related cataracts are thought to contribute most to the burden of visual impairment among leprosy affected individuals, however, the often irreversible and severe nature of the visual impairment caused by lagophthalmos makes it an important element of strategies to reduce the burden of visual impairment in leprosy affected individuals [4,5].

A recent cross-sectional study in Brazil found the prevalence of ocular abnormalities in those affected by leprosy was 100% [6]. The most prevalent findings were Meibomian gland dysfunction and dry eye syndrome, importantly they found that there was no significant difference in ocular manifestations in those beginning or who had already finished MDT, signalling the need for long term ocular management.

Lagophthalmos is the incomplete or abnormal closure of the eyelids and has many different causes, one of which being leprosy [7]. Lagophthalmos is a well-documented complication in leprosy and occurs due to the involvement of the seventh cranial nerve which causes incomplete closure of the eyelids [8]. Lagophthalmos can be either paralytic in nature or cicatricial. Leprosy causes a paralytic lagophthalmos similar to idiopathic nerve paralysis (Bell's palsy). Paralytic lagophthalmos in leprosy is thought to result from *Mycobacterium leprae* invasion on the peripheral endings of the facial nerve, and resultant immune mediated nerve-damage which can progress post microbiological cure [9]. The nerve damage causes weakness and eventual paralysis of the orbicularis muscles of the eyelids, leading to a reduced ability and eventually an inability of the lids to completely close [9]. The resultant lagophthalmos can lead to exposure keratitis and can be potentially sight threatening [10].

Lagophthalmos can be managed conservatively in its early stages using, eye taping, eye drops, punctal plugs, eyelid stretch exercises, and physiotherapy [11]. Lagophthalmos secondary to leprosy can also be treated with steroids in the early stages (recommended >6months duration of neural sequelae) due to the aetiology of facial nerve inflammation [9,12]. However, diagnosis in leprosy can often be delayed [13], this can lead to patients presenting with severe lagophthalmos at diagnosis and subsequently require surgery to prevent progression of visual disability.

To date however no systematic review has investigated the surgical management of lagophthalmos secondary to leprosy. A previous questionnaire based study published by Courtright and Lewallen [14] found "that surgeons in Asia used a wide range of different techniques for the correction of lagophthalmos while almost all of the surgeons in Africa used tarsorrhaphy" and identified that "effectiveness of the different procedures commonly used has not been measured". This review will have two main aims primarily to identify and evaluate the techniques that exist in the literature for the correction of lagophthalmos in those affected by leprosy and secondly to identify the evidence, efficacy and reported side effects in the literature for the various techniques identified.

## Methods

### Design

A systematic review, following the PRISMA guidelines [15], was carried out to synthesise evidence on the surgical correction of lagophthalmos secondary to leprosy, with the final search being completed on the 16/11/2024. The PRISMA checklist is attached in S1 Checklist: PRISMA Checklist.

### Eligibility criteria

Eligible studies were in English, there were no restrictions on geographical location. Studies were included where a surgical intervention to address lagophthalmos secondary to leprosy had been researched. There was no restriction on study design. The PICOS criteria [16] used for this study are: Population – Patients with lagophthalmos secondary to leprosy; Intervention – surgical correction of lagophthalmos; Outcome – surgical outcome post procedure; Comparison – complications, success rate between reported techniques used; and Study type –peer-reviewed journal articles, no restriction on methodology. Furthermore, grey literature manuscripts and editorials were deemed not acceptable, as well as non-peer reviewed articles. Results were required to be stratified to the particular surgical procedure(s) identified.

### Search strategy

The automated search detailed below was completed in PubMed, Web of science core collection, Medline ALL and Infolep with the string for each detailed in S1 File. The MeSh terms and free text terms for PubMed were generated based on the identified PICOS criteria.

### Data collection and management

Data were initially maintained and managed using endnote referencing software and were manually deduplicated by MW, HB, and DM. Deduplicated results of the initial search were then reviewed using Rayyan [17]. The studies remaining were subjected to abstract and title screening by MW, DM and HB using Rayyan's "blind" function, where at least one of the three authors listed the paper as "include" or "maybe" it the proceeded to full text screening. The identified studies were then reviewed independently as a full text by MW, DM, and HB before consensus was reached on inclusion/exclusion.

### Data extraction and analysis

After sorting to isolate papers to include in the review information was collated and tabulated by HB and DM with this then checked by MW and AF. Data extraction included setting, sample, study design, surgical technique, reported outcomes post-surgery, length of follow up, reported adverse effects, and other relevant data, and is summarised in S2 File. Further data for each surgical technique identified is listed in the relevant section in the results.

### Risk of bias

In keeping with PRISMA guidelines each paper was screened for independent bias; Mixed method appraisal tool (MMAT) checklists were completed by HB and DM to screen for any potential bias. A score was then calculated based on the recommendations of Pace et al. [18] and are summarised in S3 File [19].

### Results

The initial database search identified 272 potentially relevant papers. After de-duplication (removing 174 papers), title and abstract screening (removing 61 papers), 37 papers were sought for retrieval as full texts, with 29 full texts retrieved for screening. 12 papers were found eligible for inclusion in this analysis (Fig 1).

The 12 papers included contain data from seven countries, India [20–24], Korea [25,26], Nepal [27], Indonesia [28], Egypt [29], Spain [30], Thailand [31].

The majority of papers (n = 7) studied various techniques of Temporalis Muscle transfer (TMT). Gillies technique or modifications of this technique were reported in five papers [20,22,24,25,27]. Three papers reported modifications of Johnson's method of temporalis tendon transfer [21,22,27]. One paper reported TMT using a silicone sling [23]. Patients also received TMT using the Brown-McDonnell and the McCord-Cordner techniques [25]. Lagophthalmos

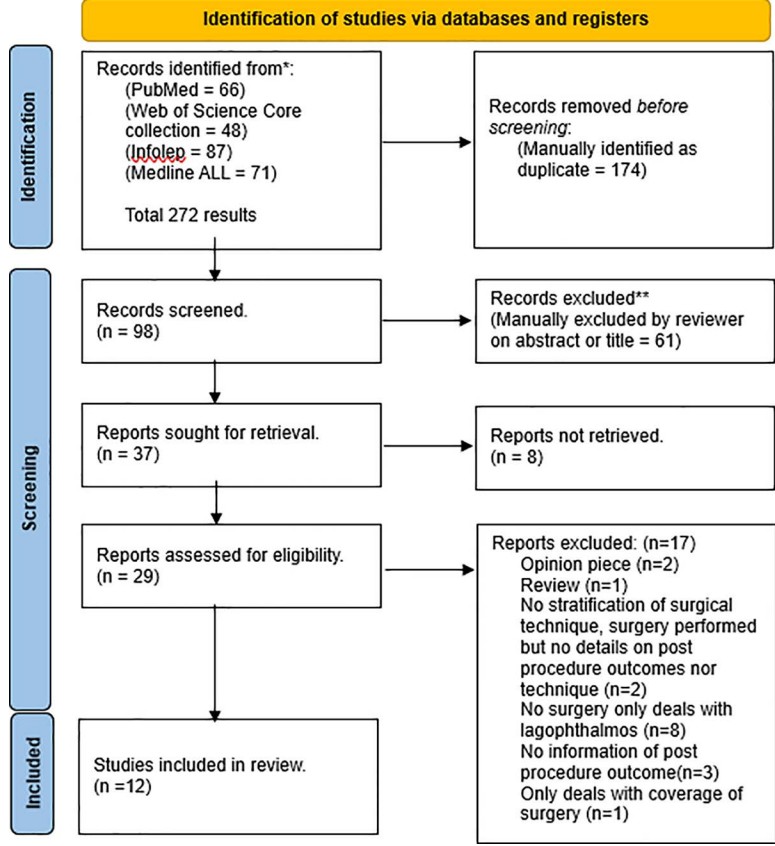

**Fig 1. PRISMA flowchart.**

was also corrected by the use of weight implant techniques, two papers studied the use of gold weight implants (GWI) [28,29] and a single paper the use of stainless steel implants (SSI) [31]. One paper each studied lateral tarsal strip [26], modified tarsorrhaphy (MT) [28] and scapha graft [30]. All surgical techniques are summarized at the end of the results section.

The majority of the included papers were prospective or retrospective studies of surgical patients, with a single randomized control trial investigating modified tarsorrhaphy versus gold weight implant [28], and a single case study [30].

## Methodological quality of results

The methodological quality of the papers included are summarized in S3 File. All papers were assessed using the MMAT. Under the definitions provided by the MMAT 11 papers were found to be quantitative descriptive studies [20–27,29–31] and one paper was identified as a quantitative randomized control trial [28].

Overall, the results found that the methodological quality of the evidence was generally quite high for each class of intervention, with 7/12 papers scoring a score of >70% for the methodological appraisal [21–23,26–29]. However, five studies the methodological quality was more modest [20,24,25,30,31].

## Surgical techniques identified

A summary of the results for each surgical procedure identified by the review is summarised below in Table 1.

**Table 1. Summary of results.**

| Proce-dure Used | Refer-ence | Year | Setting | Type of Study | Number of Eyes | Number of Patients | Outcomes | Follow Up | Complications Mentioned (incidence rate) |
|---|---|---|---|---|---|---|---|---|---|
| TMT | [18] | 1961 | India | Retrospective interventional study. | Not discussed | 10 | 8/10 patients could achieve full active eye closure bilaterally | 1-3 months | Postoperative oedema (incidence not reported) and experienced exposure symptoms (incidence not reported) |
| TMT | [19] | 2011 | India | Retrospective interventional study | 101 | 69 | 85% of the eyes could achieve full lid closure | 5 years | Exposure keratitis (16) and epiphora (31) |
| TMT | [20] | 2015 | India | Prospective interventional study. | 22 | 20 | 16 patients achieved complete eyelid closure | 6 months | Not discussed |
| TMT | [21] | 2014 | India | Prospective interventional study | Not discussed | 10 | 80% of patients achieved a lid gap of less than 1mm postoperatively | 3 months | Not discussed |
| TMT | [22] | 1984 | India | Prospective interventional study | Not Discussed | 20 | 19 cases experienced '+++ tone' postoperatively | Not discussed | Cold abscess formation at the muscle-fascia junction (1) and post-operative haemorrhage (2) |
| TMT | [23] | 2016 | Korea | Retrospective interventional study | 75 | 60 | 12 patients showed 2–3mm of lid gap postoperatively and 33 patients, had a postoperative lid gap of less than 2mm. | Not discussed | Postoperative ptosis (15.4%) |
| TMT | [25] | 1997 | Nepal | Retrospective interventional study | 51 | 35 | Preoperative lid gap on tight closure changed from 5.3mm preoperatively to 1.0mm on discharge to 0.4mm on follow-up | 7 years and 3 months | Ectropion development (6), and overtightening of the upper lid to allow for adequate opening of the eye (3). |
| Lateral Tarsal Strip | [24] | 2017 | Korea | Retrospective interventional study | 44 | 40 | 65% improvement on pre-operative symptoms | 12 months | Bilateral mild lid size discrepancy (2) and paralytic ectropium (5) |
| GWI | [26] | 2022 | Indonesia | Randomised controlled trial study | 12 | 12 | Lid gap changed from 3.21 mm preoperatively to 0.83 mm postoperatively at follow up | 1 year | Local allergic reaction to implants (incidence not reported) and implant extrusion (2) |
| GWI | [27] | 2010 | Egypt | Prospective interventional study. | Not discussed | 12 | Complete lid closure was achieved in 66.66% of patients | 3 months | Implant extrusion (12) and chronic inflammatory reaction (2). Postoperative lid oedema and/or ecchymosis (not reported). |
| SSI | [29] | 1999 | Thailand | Prospective interventional study | 22 | 20 | Complete closure or residual eyelid palpebral fissure distance of 0.5-1.5mm was obtained in 90% of cases. | 12 months | Thinning of the skin (2) and tightening of the lid (2). |
| MT | [26] | 2022 | Indonesia | Randomised controlled trial study | 11 | 11 | Lid gap changed from 3.09 mm preoperatively to 0.43 mm postoperatively at follow up | 1 year | Not discussed |
| Scapha Graft | [28] | 2009 | Spain | Case Report | 2 | 1 | Good protection of the cornea and satisfactory healing of auricular donor site was reported at follow up. | 3 years | Not discussed |

## Temporalis muscle transfer

Andersen [20] describes the use of the Gillies procedure on ten patients, all of which all had a follow-up within three months. Eight out of the ten patients could achieve full active eye closure bilaterally post operatively, with the same number able to close their eyes during sleep. The author also reports a 100% restoration rate of lacrimal duct patency postoperatively and the ceasing of exposure symptoms, in all patients, four weeks postoperatively. Regarding ocular complication, the only reported complications postoperatively were found to be oedema and exposure symptoms which were reported to subside 1 week after surgery.

Verma et al. [24], described the use of the Gillies procedure on 20 cases, suffering from either unilateral or bilateral lagophthalmos with varying degree of exposure keratitis. The study analysed the lid function and tear film morphology of each patient pre- and postoperatively. Regarding lid function, exposure keratitis was scored on a scale of nil to minimal, moderate and severe, whereas degree of epiphora, measure of orbicularis tone and degree of ectropion were all scored on a scale of +, to ++, to+++ for varying severity. The paper reports 11 patients, 2 patients and 7 patients with a minimal, moderate and severe exposure keratitis respectively preoperatively, with this changing to 19 patients with nil symptoms and 1 with mild symptoms of exposure keratitis postoperatively. The study reported 1, 3 and 16 cases of +, ++ and +++ degree of epiphora respectively preoperatively with this being converted to just 2 cases of + severity postoperatively. Regarding orbicularis tone, the study reported 1, 2 and 17 cases with +++, ++, and + measures respectively of tone preoperatively, with this being changed to 19 cases of +++ tone and 1 case of + tone postoperatively, And finally, degree of ectropion was changed from 1, 2, and 17 cases of +, ++ and +++ severity respectively preoperatively to only one case of + severity Ectropion post operatively. With respect to tear film morphology testing, the paper fails to include indication of which readings were taken pre- and postoperatively. The ocular complications of this procedure reported by the authors were summarised as the formation of a cold abscess at the muscle-fascia junction in a single patient who was suffering from pulmonary tuberculosis and two patients who suffered from postoperative haemorrhage.

Ahn et al. [25], discussed the use of the Gillies procedure in 15 patients, 8 using the Gillies-Andersen method and 7 using the modified Gillies method. The authors described no incidence of postoperative ptosis using the modified Gillies-Andersen method, whereas the modified Gillies method resulted in an incidence of postoperative ptosis of 15.4%. No other post operative complications were mentioned in this study.

Das et al. [21], reported solely on the use of the Johnson's method for correction of lagophthalmos in leprosy patients. The authors reported on data from 69 patients over an 11-year period. The authors report that the average lid gap on gentle closure changed from 7.9mm preoperatively to 2.4mm postoperatively and to 1.9mm at 5-year follow-up. In the same timeframes, the lid gap on forced closure were reported as, 4.8mm, 0.2mm and 0.3mm. This distance between lids on straight gaze showed the change to be from 12.6mm preoperatively, to 9.8mm postoperatively to 9.9 after 5-year follow-up. The authors reported that 85% of the eyes could achieve full lid closure after receiving a course of postoperative physiotherapy. Regarding postoperative ocular complications, that authors described reducing the occurrence of exposure keratitis in 16 patients, mild epiphora in 28 patients, moderate epiphora in 2 patients and severe epiphora in 1 patient. No comment was made on whether patients experienced any other form of postoperative complication.

Two authors [22,27] also discuss the use of the Gillies procedure alongside discussion of the Johnson's procedure however, no distinction within the results section is used to distinguish the results of each respective method.

Singhal et al. [22], describe using the Gillies procedure in 13 cases and the use of the Johnson's procedure in 7 cases with a 6-month minimum follow-up period. The study discusses the functional changes in mean lid gap both pre- and postoperatively (at 6 month follow-up) during straight gaze, gentle closure and forced closure: At straight gaze, the mean lid gap was reduced from 12mm to 9.09mm; at Gentle closure the mean lid gap was changed from 7.81mm to 1.59mm; and at forced closure, the lid gap was reduced from 4.13mm to 0.50mm by the 6 month follow-up period. The study also measured qualitative measures of patient satisfaction post operatively: the study reported that,16 patients reported "excellent results" reporting being able to achieve complete eyelid closure with chewing; "good result(s)" were also reported as

eyelid closure (within 1–2mm) while chewing was found in 4 cases patients and a "fair" result was described as incomplete eyelid closure alongside chewing with compromised corneal protection which was found in 1 patient postoperatively. Postoperative complications were not discussed within this paper.

Soares and Chew [27] also discussed the use of the Johnson's procedure alongside the use of the Gillies procedure with no clear distinction made between the 2 techniques within the results. This paper discussed the use of the Johnosn's procedure in 47 cases and the use of the Gillies in 4 cases. The average length of discharge was reported at 60 days and follow-up was reported to be 7 years and 3 months by the authors. The authors report finding "no significant difference between preoperative and follow-up measurements of orbicularis oculi muscle strength and corneal sensation". The authors report finding that preoperative lid gap on light closure changed from 7.3mm to 3mm on discharge and again to 3.2mm on follow-up. Using the same timelines the lid gap on tight closure changed from 5.3mm to 1.0mm to 0.4mm and the blink frequency changed from 6.7/min to 9.2/min to 7.0/min. Corneal anaesthesia was reported to be found in 31% of patients preoperatively, 39% at discharge and 30% at follow-up. Regarding postoperative complications, the authors reported that in 6 eyes an ectropion developed, and in 3 cases, the upper lid was reportedly too tight to allow for adequate opening of the eye, the average reporting incidence of this was recorded to be found at 3 months.

Gupta et al. [23], were the only authors to comment on the use of a silicone sling to perform a modified TMT. The authors describe the use of this method on 10 patients. The authors report that 80% of patients a lid gap of less than 1mm was achieved post operatively. The authors report that the average preoperative lid gap, on eye closure changed from 7.7mm to 0.5mm on day 1 postoperatively and 0.70mm 3 months postoperatively. A significant reduction in palpebral aperture was reported postoperatively. The authors report the average preoperative vertical palpebral aperture to be 12.05mm, reducing to 10mm on day 1 postoperatively and 10.35mm 3 months postoperatively. No postoperative ocular complications were mentioned by the authors.

Ahn et al. [25], were the only authors found by this review to discuss the use of the Brown-McDowel method of TMT. The authors report the use of this method in 12 patients describing the procedure to show 2–3mm of lid gap postoperatively. No postoperative ocular complications were noted by the authors.

Ahn et al. [25], were also the only authors found by this review to discuss the use of the McCord-Cordner method of TMT. The authors reported the use of this procedure in 33 patients, achieving a postoperative lid gap of less than 2mm. It was reported that postoperative ptosis was found in this cohort to be at an incidence of 21.4%.

## Weight implantation

Weight implantation was studied by 3 papers [28,29,31]. One paper studied GWI [29], One paper SSI [31] and one paper was an RCT comparing GWI and MT [28].

## Gold

GWI was researched by El Toukhy [29] in a prospective study of 12 leprosy patients in Cairo, Egypt. This study included 12 male leprosy patients with an average age of 46. Seven of the included patients had been treated as paucibacillary and 5 as multibacillary. At 3 months follow-up, 11 patients (91.66% of patients) had satisfactory closure (defined as a reduction of the lid gap of 3 mm or more). Complete lid closure was achieved in eight out of the 12 cases (66.66% of patients). Incomplete closure was attained in four patients; however, they reported that these four patients still attained a decrease in corneal irritation.

Six out of 12 implants were extruded within the first year. Two more implants had to be removed due to chronic inflammatory reaction. This lowered the success rate to 33% (4/12 cases) after 1 year. Minor transient complications in the form of postoperative lid oedema and/or ecchymosis which resolved spontaneously were reported as being present though incidence was not recorded.

## Stainless steel

SSI was investigated by Kuntheset [31] in a prospective study of 20 Thai patients with paralytic lagophthalmos, 18 of whom had lagophthalmos secondary to leprosy. Only one eye was operated on in each of 18 patients and both eyes in two patients were operated on for a total of 22 eyes. The patients ranged in age from 29 to 65. The surgery was graded as excellent if complete closure was obtained, moderate if a residual eyelid palpebral fissure distance of 0.5-1.5 mm persisted. Both excellent and moderate results are accepted as a success, and the success rate is, therefore, about 90% in this study. After more than 12 months of follow up, 20 of the 22 eyes were still in good condition.

In two eyes after 7–8 months, the plates were removed due to thinning of the skin and tightening of the lid. Postoperative complications, such as ectropion, entropion and overtightening, were not found in this study. They reported the fact that SSI were cheaper than GWI.

## Gold weight implant vs Modified Tarsorrhaphy

Irawati et al. [28] studied GWI against MT in a one year observational randomized control trial. The study consisted of 11 eyes in the MT group (intervention group) and 12 eyes in the GWI group (control group). 18 patients (78.3%) were male and 5 (21.7%) were female. Almost all patients had MB-type leprosy (21 patients or 91.3%).

Lagophthalmos distance decreased in the MT (3.09 mm to 0.43 mm) and GWI groups (3.21 mm to 0.83 mm) at postoperative year 1. The MT and GWI techniques showed no significant difference in decreasing lagophthalmos distances with or without gentle pressure at nasal, central, and temporal areas at 1 year, with the difference only being significant at 3 months ($p = 0.044$). Ocular Surface Disease Index score, tear break-up time, and Schirmer test without and with anesthesia in the MT and GWI groups showed a p-value of $> 0.05$. Corneal sensitivity changes in the inferior quadrant of the MT group (50.00 to 51.30 mm) and in the GWI group (49.61 to 52.93 mm) resulted in a $p > 0.05$.

No major complications were found in the MT group, while two eyes (15%) in the GWI group experienced implant extrusion. Patients in GWI group experienced limited visual fields when the eyes were open due to the weight in their upper lids. A commonly reported side effect of GWI was a local allergic reaction to the gold implant.

Surgical duration for those in the MT group (44,61 ± 11,29 min) was insignificant when compared to those in the GWI group (43,81 ± 15,03 min). In terms of cost, the mean cost of surgery in the MT group was equal to that of the GWI group. Only two patients who had implant extrusion underwent the MT technique as a reparative procedure. The final mean cost of those in the GWI group after complication correction was significantly higher (3.017.437,54 ± 560.823,97 IDR) than of those in the MT group ($p < 0.05$).

## Other techniques

**Lateral tarsal strip.** Jue et al. [26], were the only authors identified by this review to discuss the use of lateral tarsal strip for surgical correction. The authors described the use of the procedure in 40 patients. A 4-point patient's global assessment scale (range 0–4) was also used which collected information on symptomatic improvement postoperatively, of which the mean score was reported at 2.6/3 at the end of the 12-month follow-up period. This score indicated roughly a 65% improvement on preoperative symptoms such as the patient's ability to close the eyelids, redness, discharge, epiphora, and sensation of foreign body. The authors describe no serious postoperative ocular complications except for the occurrence of a bilateral mild lid size discrepancy in 2 patients of about 1–2 mm and the presence of postoperative recurrence of the paralytic ectropium was observed in 5 patients.

**Scapha graft.** One case report from Spain detailed the use of a scapha graft in a 36-year-old Mozambican man to treat bilateral lagophthalmos that was refractory to a lateral tarsal strip procedure [30]. They report good protection of the cornea and satisfactory healing of auricular donor site at 3 year follow up. No post operative complications were reported.

## Summary of surgical techniques

The surgical techniques included in this review are summarised below in Table 2.

## Discussion

This review has identified 5 broad techniques for the surgical correction of lagophthalmos secondary to leprosy, these include: Temporalis muscle transfer; Weight implantation techniques; Lateral tarsal strip; Modified Tarsorrhaphy; and

**Table 2. Surgical techniques for the correction of lagophthalmos secondary to leprosy.**

| Surgical Technique | Description |
|---|---|
| Temporalis Muscle Transfer (TMT) | The procedure is done under local or general anaesthesia, and the area of skin incision is shaved prior to the procedure. A hook-shaped skin incision is created, exposing the temporalis fascia from its origin to insertion. A 1 cm wide strip of fascia is isolated and carefully separated from the muscle using a knife. The central strip of the temporalis muscle, along with the over-lying fascia, is detached from its origin and the underlying temporal bone, then reflected downward toward the mandible. The lower lid fascia strip is passed through the passage below the ligament in the medial canthal area. The tip of the strip passed is sutured in loop shape under high tension and fixed to the palpebral ligament and near nasal periosteum. The fascia slip in the upper lid is not joined with that of the lower lid at the medial canthal tendon to lower tension. The skin incision is sutured closed and a pressure dressing applied over the temple. All dressings are then opened after 24 hours. There are numerous variations of this surgical method which include Brown-McDowell method, McCord-Codner method, Gillies-Andersen method, modified Gillies-Andersen method, and Johnson's method. – adapted from: **Temporalis Muscle Transfer for the Treatment of Lagophthalmos in Patients with Leprosy Refinement in Surgical Techniques to Prevent Postoperative Ptosis [22]** and **An evaluation of Gillies' procedure for lagophthalmos in leprosy [21]** |
| Modified tarsorrhaphy | This is carried out in three steps (Yunia technique). Levator recession is performed first. This involves a local anaesthesia injection into the upper eyelid and lateral side skin crease incision and orbicularis dissection up to the tarsal plate. Then the conjunctiva is everted with ballooning of the levator recess followed by a lid crease suture. After this lateral tarsorrhaphy is performed. This involves canthotomy, lateral cantholysis, and excision of the upper and lower lid margins, followed by permanent lateral tarsorrhaphy. Finally, canthopexy and canthoplasty or lateral tarsal strip and canthoplasty are performed according to the horizontal eyelid laxity. The skin is then sutured. – adapted from: **Modified tarsorrhaphy versus gold weight implant technique for paralytic lagophthalmos treatment in patients with leprosy: One-year observation of a randomized controlled trial study [25]** |
| Scapha graft | The conjunctiva is initially dissected through a subtarsal incision, and the lower lid retractor layer is separated from the lower border of the tarsal plate before being dissected from the underlying orbicularis muscle until retractors are released. Cartilage and perichondrium are then removed by an elliptical incision of the scapha. The graft is then cut and moulded to fit the ocular globe. It is then sutured to the tarsal plate to support the lower lid. The lower portion of the graft is sutured to the retractors and then brought out through the skin. The cartilage is covered with the previously dissected conjunctiva and then a Frost stitch is used for 7 days after the application of antibiotic ointment to the cornea. - Adapted from: **Surgical treatment of bilateral paralytic lagophthalmos using scapha graft in a case of lepromatous leprosy [27]** |
| Lateral tarsal strip | This surgical technique involves creating a lateral canthotomy that extends to the lateral orbital rim. A lateral cantholysis is performed by incising the inferior crus of the lateral canthal tendon. The eyelid is then divided into anterior and posterior lamellae, and a tarsal strip is prepared by trimming the mucocutaneous junction superiorly. Once the tarsal strip is fashioned, it is shortened, lifted with forceps, and the conjunctival surface is scraped. The prepared tarsal strip is then sutured to the periosteum on the inner surface of the lateral orbital rim, above the insertion of the lateral canthal tendon. Finally, the lateral canthotomy is closed after removal of any redundant skin. – adapted from: **The Lateral Tarsal Strip for Paralytic Ectropion in Patients with Leprosy [23]** |
| Gold weight implant | Gold weight implantation is performed using local anaesthesia injected into the upper eyelid. A skin crease incision is then made, followed by dissection of the orbicularis muscle. Following this, the orbital septum is opened, and the conjunctiva is everted and ballooned. The gold weight is then placed inferior to the levator insertion on the tarsal plate and sutured. Then the pretarsal and preseptal orbicularis muscles are sutured to cover the implant before then finally suturing the skin. - adapted from: **Modified tarsorrhaphy versus gold weight implant technique for paralytic lagophthalmos treatment in patients with leprosy: One-year observation of a randomized controlled trial study [25]** |
| Stainless steel weight implantation | Surgery is performed under local anaesthesia injected into the upper eyelid. A 15 Bard-Parker blade is used to incise the skin and Westcott scissors are used to enter the elevated orbicular plane. A stainless-steel plate is inserted into the pocket between the orbicularis muscle and the orbital septum-tarsal plate which is then set over the medial two thirds of the superior tarsal border. The lower part of the pocket is sutured to the superior tarsal border and the upper part is sutured to the orbital septum. The overlying orbicular muscle and skin is then sutured together. – adapted from - **Reanimation of the lagophthalmos using stainless steel weight implantation; a new approach and prospective evaluation [28]** |

Scapha grafting. In this discussion we will be analyzing each of these broad techniques to gain a deeper understanding of their success and drawbacks, particularly in the context of leprosy treatment.

## Overall postoperative results and complications of surgical modalities

TMT was identified most commonly by this review and all results identified were shown to have a positive outcome postoperatively. All papers described a positive improvement to the functional and, when reported, aesthetic metrics and patient descriptions. 5 studies reported that TMT led to achieving complete eyelid closure in 80–85% of patients [20–23] with one study stating complete eyelid closure was reported in 95% of patients [24]. One study stated that postoperatively, 80% of patients described "excellent results" [22]. The technique was shown to have positive impacts on the patency of the lacrimal duct [20], reduction of exposure keratitis experienced by patients and removal of preoperative ectropium [24]. However, this method is not without its ocular complications. Studies reported a variety of ocular complications when using this method of surgical correction. These were reported from more minor, spontaneously resolving, ocular complications such as, postoperative oedema formation [20] and exposure keratitis [20,21], to more serious ocular complications such as, cold abscess formation at the muscle fascia junction [24], post operative ptosis [25], epiphora [21], ectropion formation [27], and finally one patient was reported to be unable to open their eye postoperatively due to extreme tightness of the upper eyelid [27].

Weight Implantation was discussed by 3 papers in this review. The success of this technique was reported to be more varied, however. The technique was reported to be highly effective immediately postoperatively, with one study reporting a success rate of 90.9% [31] and another reporting a mean lid gap distance of 0.83mm 1 year postoperatively [28]. This technique was also reported to be effective in improving lacrimation for patients postoperatively. Complete lid closure, however, was reported as more variable amongst those undergoing surgery, with one study achieving 66.7% complete lid closure [29], and another only reporting only a 36.6% complete lid closure rate [31]. However, one crucial risk of this technique was found to be the high rate of extrusion of the implant and medically indicated removal postoperatively. Both GWI studies reported extrusion of the implant, one study reported a 50% rate of extrusion [29] and the other reported a 15% extrusion rate [28]. Both SSI and GWI studies reported requiring medically indicated removal of the implants postoperatively, with one study reporting a 16.7% medically indicated removal rate postoperatively [29] and another study reporting that 10% of patients required removal of the implants 7–8 months postoperatively [31]. Reasons for medically indicated removal include, thinning of the skin and tightening of the lid and chronic inflammatory reaction to the implanted weights. Extrusion was only reported in gold weight implantation, however, medically indicated removal of the weights was reported in both gold and stainless-steel implantation. Ocular complications were only reported in relation to gold weight implantation, and these range from general ocular complications such as post operative oedema and ecchymosis, to larger impact ocular complications such as reduction in field of vision due to the increased weight in the upper eyelids and local allergic reaction to the implant [28,29]. It has been advised that an allergy check is necessary before conducting this procedure in patients with a previous history of gold or metal allergy to avoid allergic reaction [28] and in those who experience reaction to GWIs, SSIs may be offered in place.

Modified tarsorrhaphy, Lateral tarsal strip and Scapha grafting were all only reported by one paper in this study. Modified Tarsorrhaphy was reported to have a very positive impact on reducing the lagophthalmos distance to only 0.43mm 1 year post operatively [28]. No major ocular complications were mentioned in relation to the surgical procedure however, this procedure was also shown to be effective in improving lacrimation for patients postoperatively. This technique was directly compared to GWI as part of a RCT however, no statistically significant difference was observed between the two surgical modalities [28].

Lateral tarsal strip surgery was reported to show about a 65% improvement on preoperative symptoms experienced by patients [26]. This surgical modality was reported to have little postoperative complications except for the occurrence of a bilateral mild lid size discrepancy in 2 patients of about 1–2 mm [26]. Moreover, Kopecký et al. [32], discuss that the main advantage of this technique is the ability to properly reposition the eyelid and to allow for rejuvenation of its lateral fixation.

Scapha grafting was reported to be a successful treatment for a single leprosy patient with lagophthalmos refractory to MT which was detailed to show long lasting results at 3 a year follow-up period. No post operative complications were reported by the report [30].

## Consideration of treatment in the setting of Leprosy

Given the low-resource context in which many leprosy affected individuals live, it is hugely important to recontextualize their effectiveness based upon the resources available within the region of use. Within the techniques a range of aesthetic and functional outcomes are noted. Where patients are able to be followed up and resources allow, more resource intense but functionally and aesthetically preferrable procedures such as TMT may be preferred. However, in reduced resource settings factors such as: the type of anaesthesia required; level of expertise needed to conduct the operation; success rate; incidence and risk of complications; and longevity and stability of the results are vital to consider [33,34]. As leprosy is a condition primarily affecting low- and middle-income countries [35], these factors play a large role in determining the success and outcomes of a case [33]. It has been shown that in low- and middle-income countries that lack of adequately skilled staff and healthcare worker shortage leads to a reduction in availability of surgical treatment. Thus, the requirement for a trained anaesthetist to be present at the operation may serve as a barrier to treatment for those needing to undergo surgical correction of lagophthalmos. With the exception of TMT, all identified procedures in this review are predominantly carried out under local anaesthetic [25,28,30,32], this proves to be extremely advantageous within the context of surgical eye-camps where many of these procedures are performed in the field [34]. Furthermore, lower resource settings my not have the ability to conduct repeat operations to correct failure or ocular complications as a result of a surgery, thus, operations which lend themselves to be effective long term and with a low failure rates will be preferential within these settings particularly. Therefore, procedures such as weight implantation, with a proportionally larger complication rates may not be as favourable in low-resource settings as other methods identified within this review. Moreover, the cost per patient is another consideration within the lower resource setting [33] as for example SSIs are cheaper than GWIs and MT was found to be much cheaper than GWI [28,31] this may act as a further financial incentive for eye care professionals working in lower resource settings.

## Limitations and strengths

This review covers all publication dates of evidence available for the surgical correction of lagophthalmos secondary to leprosy and is the first study to systematically evaluate all evidence available in the literature. However, potentially relevant papers may have been missed due to not being included in the databases searched, being published in a language other than English, or by not being available as a full text for evaluation. Due to the nature of the data presented and the varying outcome measures in the included studies it was also not possible to perform a meta-analysis of the pooled results.

## Conclusion and recommendations

As many patients who suffer from lagophthalmos secondary to leprosy are found in poor, remote, and under-resourced communities the use of the surgical procedures outlined will face many practical barriers. The favouring of techniques shown to require less resources and that require less follow-up such as MT may be favoured in these settings compare to TMT which often requires more training for surgeons and GWI which has a high rate of extrusion and need for follow-up. However, a convincing argument has emerged that lagophthalmos and visual impairment secondary to leprosy have been ignored with only 12 papers found by this review. Even more importantly the paucity of well-designed comparison studies between procedures, particularly in low-resource and remote settings highlights the need for future studies to compare the effectiveness and practicality of the identified techniques in endemic regions. Future studies into the management of

lagophthalmos will help policy makers re-evaluate the best surgical technique for the individual patients and communities they serve, not only improving quality of life but avoiding the progression of a potentially treatable cause of blindness.

## Supporting information

**S1 Checklist.  PRISMA checklist.**
(DOCX)

**S1 File.  Search strands.**
(DOCX)

**S2 File.  Characteristics of included studies.**
(DOCX)

**S3 File.  Risk of bias assessment.**
(XLSX)

## Author contributions

**Conceptualization:** Matthew Willis, Heynes Brown, Anil Fastenau.

**Data curation:** Matthew Willis, Heynes Brown, Dan McGrath.

**Formal analysis:** Matthew Willis, Heynes Brown.

**Investigation:** Matthew Willis, Heynes Brown, Dan McGrath.

**Methodology:** Matthew Willis, Heynes Brown, Essam Eltoukhy.

**Supervision:** Anil Fastenau.

**Validation:** Matthew Willis, Heynes Brown.

**Visualization:** Matthew Willis, Heynes Brown.

**Writing – original draft:** Matthew Willis, Heynes Brown, Dan McGrath, Essam Eltoukhy, Anil Fastenau.

**Writing – review & editing:** Matthew Willis, Heynes Brown, Dan McGrath, Essam Eltoukhy, Anil Fastenau.

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
