## [Decision Letter · Decision Letter 0]

Response to Reviewers
Revised Manuscript with Track Changes
Manuscript

Shaden Kamhawi

co-Editor-in-Chief

Paul Brindley

co-Editor-in-Chief

**Additional Editor Comments (if provided):**
**Journal Requirements:**
**Reviewers' comments:**

**Key Review Criteria Required for Acceptance?**

**Methods**

-Are the objectives of the study clearly articulated with a clear testable hypothesis stated?

-Is the study design appropriate to address the stated objectives?

-Is the population clearly described and appropriate for the hypothesis being tested?

-Is the sample size sufficient to ensure adequate power to address the hypothesis being tested?

-Were correct statistical analysis used to support conclusions?

-Are there concerns about ethical or regulatory requirements being met?

Reviewer #1: Study design is appropriate, but available literatures are limited

Most are retrospective . study. One case report Scapha grafting may not appropriate to include in this revision.

Reviewer #2: • What experiments or interventions were used? NA

• Are the experiments or interventions appropriate for addressing the research question? NA

• Are conditions adequate and the right controls in place? NA

• Is there enough data to draw a conclusion? 12 articles were selected; in this review this seems adequate. Perhaps French literature would have provided extra info?

• Do the authors address any possible limitations of the research? NO, but not needed

• Was data collected and interpreted accurately? Yes, in a very accurate way!

• Do the authors follow best practices for reporting? Yes

• Does the study conform to ethical guidelines? NA

• Could another researcher reproduce the study with the same methods?

In other words, have the authors provided enough information to validate

the study? YES

Reviewer #3: The objectives are clearly written, study design is also well written.

Based on prisma checklist, did you use any method to assess certainty (or confidence) in the body of evidence for an outcome?

**Results**

-Does the analysis presented match the analysis plan?

-Are the results clearly and completely presented?

-Are the figures (Tables, Images) of sufficient quality for clarity?

Reviewer #1: Although enough, but Technique of operations and complication in table or image will be helpful for better understanding

Reviewer #2: • Do the results support the conclusions? YES

• Do the conclusions overreach? NO

• Do the authors discuss any limitations of the study? YES

• If the journal selects based on advance in the field does the study demonstrate this advance? IT IS A GOOD REVIEW, ANY ADVANCE IS NOT DISCUSSED.

Reviewer #3: The analysis prsented match the analysis plan and clearly presented. However maybe It might be helpful to include a brief explanation regarding potential differences in GWI usage across cases or groups. Additionally, consider discussing whether there is any observed correlation between GWI usage and inflammation, particularly in relation to the possibility of extrusion due to foreign body response.

Also for the table, i think it would be good to add extra rows for success rate for each methods, patient characteristics, complications for each study.

**Conclusions**

-Are the conclusions supported by the data presented?

-Are the limitations of analysis clearly described?

-Do the authors discuss how these data can be helpful to advance our understanding of the topic under study?

-Is public health relevance addressed?

Reviewer #1: As there were limited number of publications, conclusion drawn from review is clear.

Reviewer #2: (No Response)

Reviewer #3: The conclusion is well supported by data presented. Are there any more limitations on this study? Please elaborate more on why meta analysis couldnt be performed. Also highlight on how choosing the most appropriate surgery technique is important to achieve the best result functionally and aesthetically

**Editorial and Data Presentation Modifications?**

Reviewer #1: If there will be table regarding Technique of operation ,retrospective/prospective , number of cases, complications, outcome , may be more helpful for readers

Reviewer #2: (No Response)

Reviewer #3: Minor revision

**Summary and General Comments**

Reviewer #1: Overall it is a good systematic review based on available publications. Minor revision is needed.

Reviewer #2: Some personal notes:

When faced with lagopthalmus ( in general caused by leprosy neuritis) we start with an analysis of the degree of the lid gap when the eyes are gently closed and with maximal strength, and signs of corneal damage due to dryness/infection.

When the lid gap is max 3 mm and correction is indicated I would recommend a simple medial canthoplast.

When the gap is 3-6 mm, in general a lateral suspension canthoplasty perhaps in combination with a medial suspension canthoplasty is performed. In this a suture Nylon 5x0 is fixed to the periosteum and woven through the lower orbicularis muscle and back again, and knotted. When doing only a lateral suspension one should not put any lateral stretch to the entry of the lacrimal duct.

When the lidgap is >6mm, a TMT is indicated.

You make a very valid point that too often there is little expertise and when local colleagues want to be trained one may want to create a routine in learning a not too complex technique. For this reason we often opt for a simple canthoplasty or suspension plasties as these are easier to master, do not ask for post-op physiotherapy, easier follow up, and are thus more affordable that TMT’s (or facial sling procedures) . Another important limitation is often poverty and therefore no money for transport, and the absence of a health insurance which makes day care mandatory.

In this it is after the operation accepted that often a remaining lid gap of a 2 ? mm is accepted as long as the cornea is well covered and protected.

Reviewer #3: Overall, the paper is well written

PLOS authors have the option to publish the peer review history of their article (what does this mean? ). If published, this will include your full peer review and any attached files.

**Do you want your identity to be public for this peer review?** For information about this choice, including consent withdrawal, please see our Privacy Policy .

Reviewer #1: **Yes: ** Dr Mahesh Shah

Reviewer #2: **Yes: ** Willem Theuvenet

Reviewer #3: **Yes: ** Yunia Irawati

**Figure resubmission:****Reproducibility:** To enhance the reproducibility of your results, we recommend that authors of applicable studies deposit laboratory protocols in protocols.io, where a protocol can be assigned its own identifier (DOI) such that it can be cited independently in the future. Additionally, PLOS ONE offers an option to publish peer-reviewed clinical study protocols. Read more information on sharing protocols at https://plos.org/protocols?utm_medium=editorial-email&utm_source=authorletters&utm_campaign=protocols

---

## [Decision Letter · Decision Letter 1]

Dear Mr Willis,

We are pleased to inform you that your manuscript 'Techniques for the surgical correction of lagophthalmos secondary to leprosy: a systematic review' has been provisionally accepted for publication in PLOS Neglected Tropical Diseases.

Best regards,

Susilene Maria Tonelli Nardi, Ph.D

Academic Editor

Elsio Wunder Jr

Section Editor

Shaden Kamhawi

co-Editor-in-Chief

Paul Brindley

co-Editor-in-Chief

Reviewer's Responses to Questions

**Key Review Criteria Required for Acceptance?**

**Methods**

-Are the objectives of the study clearly articulated with a clear testable hypothesis stated?

-Is the study design appropriate to address the stated objectives?

-Is the population clearly described and appropriate for the hypothesis being tested?

-Is the sample size sufficient to ensure adequate power to address the hypothesis being tested?

-Were correct statistical analysis used to support conclusions?

-Are there concerns about ethical or regulatory requirements being met?

Reviewer #1: It will be a very relevant publication in leprosy with eye problem, especially will be helpful for prevention of blindness due to leprosy.

Reviewer #3: The authors have responded accordingly to the previous review

**Results**

-Does the analysis presented match the analysis plan?

-Are the results clearly and completely presented?

-Are the figures (Tables, Images) of sufficient quality for clarity?

Reviewer #1: proper analysis of the problem based enough tables

Reviewer #3: The authors have responded accordingly to the previous review and provided additional materials

**Conclusions**

-Are the conclusions supported by the data presented?

-Are the limitations of analysis clearly described?

-Do the authors discuss how these data can be helpful to advance our understanding of the topic under study?

-Is public health relevance addressed?

Reviewer #1: conclusion is clear. It will be helpful in low recourses setting

Reviewer #3: The authors have responded accordingly to the previous review

**Editorial and Data Presentation Modifications?**

Reviewer #1: Accept.

Reviewer #3: Accept

**Summary and General Comments**

Reviewer #1: well written.

Reviewer #3: Paper is well written

PLOS authors have the option to publish the peer review history of their article (what does this mean? ). If published, this will include your full peer review and any attached files.

**Do you want your identity to be public for this peer review?** For information about this choice, including consent withdrawal, please see our Privacy Policy .

Reviewer #1: **Yes: ** Dr Mahesh Shah

Reviewer #3: No

---

## [Editor Report · Acceptance letter]

Dear Mr Willis,

We are delighted to inform you that your manuscript, "Techniques for the surgical correction of lagophthalmos secondary to leprosy: a systematic review," has been formally accepted for publication in PLOS Neglected Tropical Diseases.

Best regards,

Shaden Kamhawi

co-Editor-in-Chief

Paul Brindley

co-Editor-in-Chief
